# The accessory helix of complexin functions by stabilizing central helix secondary structure

Daniel T Radoff[1], Yongming Dong[2], David Snead[1], Jihong Bai[2], David Eliezer[1], Jeremy S Dittman[1]*

[1]Department of Biochemistry, Weill Cornell Medical College, New York, United States; [2]Division of Basic Sciences, Fred Hutchinson Cancer Research Center, Seattle, United States

**Abstract** The presynaptic protein complexin (CPX) is a critical regulator of synaptic vesicle fusion, but the mechanisms underlying its regulatory effects are not well understood. Its highly conserved central helix (CH) directly binds the ternary SNARE complex and is required for all known CPX functions. The adjacent accessory helix (AH) is not conserved despite also playing an important role in CPX function, and numerous models for its mechanism have been proposed. We examined the impact of AH mutations and chimeras on CPX function in vivo and in vitro using *C. elegans*. The mouse AH fully restored function when substituted into worm CPX suggesting its mechanism is evolutionarily conserved. CPX inhibitory function was impaired when helix propagation into the CH was disrupted whereas replacing the AH with a non-native helical sequence restored CPX function. We propose that the AH operates by stabilizing CH secondary structure rather than through protein or lipid interactions.

*For correspondence: jed2019@med.cornell.edu

Competing interests: The authors declare that no competing interests exist.

## Introduction

Precise control of synaptic vesicle fusion at the presynaptic terminal endows a nervous system with the means to regulate functional synaptic connectivity. Although many of the molecules required for neurotransmitter exocytosis are known, the mechanisms by which this process is regulated are less well understood. The core machinery of the fusion process is composed of the neuronal SNAREs (VAMP2, syntaxin 1, and SNAP-25) along with several SNARE-binding proteins such as synaptotagmin, Munc13, and Munc18 (*Sudhof and Rothman, 2009*; *Sudhof, 2013*). These proteins are highly similar in sequence and function across species, reflecting the deep conservation of synaptic transmission. Complexin is another essential SNARE-binding protein, and genetic ablation of complexin profoundly impacts neurotransmitter release and its regulation in all synaptic preparations studied to date (*Reim et al., 2001*; *Huntwork and Littleton, 2007*; *Xue et al., 2008*; *Hobson et al., 2011*; *Martin et al., 2011*; *Lin et al., 2013*; *Vaithianathan et al., 2013*). Mammals possess four isoforms of complexin, and deletion of the two broadly expressed isoforms (mCpx1/2) is lethal (*Reim et al., 2001*; *Xue et al., 2008*). This small cytoplasmic protein possesses an alpha-helical SNARE-binding domain known as the central helix (CH), which is broadly conserved among metazoa (*Pabst et al., 2000*; *Reim et al., 2001*; *Bracher et al., 2002*; *Chen et al., 2002*; *Brose, 2008*). The CH domain of mCpx1 is flanked on its N-terminus by a stable alpha helix called the accessory helix (AH) (*Pabst et al., 2000*; *Chen et al., 2002*), and this domain has subsequently been shown to play an inhibitory role in mammalian, fly, and nematode synapses (*Xue et al., 2007*, *2009*; *Yang et al., 2010*; *Martin et al., 2011*; *Cho et al., 2014*; *Trimbuch et al., 2014*). However, the primary sequence of the AH domain is poorly conserved, and its secondary structure has only been investigated in rodent complexin. A wide variety of models for AH function have been proposed including direct binding to SNAREs or other proteins

**eLife digest** The nervous system sends information around the body in the form of electrical signals that travel through cells called neurons. These signals cannot pass across the small gaps—called synapses—that separate neighboring neurons. Instead, when electrical signals reach the synapse, chemicals called neurotransmitters are released across the gap and trigger an electrical signal in the next neuron.

Neurotransmitters are stored within neurons in small envelopes of membrane known as synaptic vesicles. They are released when the vesicles fuse with the membrane that surrounds the neuron. This fusion process must be tightly controlled to ensure that information is passed between the neurons at the right time.

Complexin is a small protein that controls vesicle fusion by binding to a group of proteins called the SNARE complex. It contains two structured sections called the central helix and the accessory helix, which are both important for vesicle fusion. The central helix is able to bind to the SNARE proteins, and it has the same sequence of amino acids—the building blocks of proteins—in all animals. However, the sequence of amino acids in the accessory helix varies widely across different animals and it is not clear whether it performs the same role in all of them.

Radoff et al. studied complexin in the nematode worm *C. elegans*, and found that when its accessory helix is replaced with the amino acid sequence from the mouse one, it can still properly control vesicle fusion. Indeed, complexin can still work properly when its accessory helix is replaced with an artificial protein helix that has a similar shape.

These experiments suggest that the overall structure of the accessory helix is more important than its exact sequence of amino acids. Radoff et al. propose that its role in vesicle fusion is to stabilize the structure of the central helix to allow it to bind to the SNARE proteins. The next challenge is to understand how vesicle fusion is prevented when complexin binds to the SNARE proteins.

(*Giraudo et al., 2009*; *Lu et al., 2010*; *Yang et al., 2010*; *Kummel et al., 2011*; *Bykhovskaia et al., 2013*; *Cho et al., 2014*), electrostatic interactions with membranes (*Trimbuch et al., 2014*), and direct effects on the CH and its SNARE binding through secondary structure interactions (*Chen et al., 2002*). Do all of these mechanisms contribute to AH domain function? If the AH domain binds specifically to another protein, why is it so poorly conserved relative to the CH domain?

To investigate the mechanism of AH action and its conservation across phylogeny, we examined the AH domain structure and function in the *Caenorhabditis elegans* mCpx1/2 ortholog CPX-1 using both in vitro and in vivo approaches. While worm and mouse AH domains are two of the most divergent among published complexin sequences, the two domains could be exchanged without impairing function in vivo. Further, the recombinant worm AH formed a highly stable alpha helix in solution similar to the mouse AH. Abolishing the hydrophobic character of the worm or mouse AH domain had little effect, whereas disrupting helix stability and invasion of helical structure into the CH severely impaired complexin inhibitory function. Moreover, replacing the AH with an artificial helical sequence fully restored inhibitory function. Remarkably, this sequence was functional despite large differences in length, charge, and hydrophobicity, indicating that these properties are not critical for AH function. These experiments indicate that the principal role of the AH domain is to nucleate and propagate helical structure into the CH domain, and this function is conserved across evolution.

## Results

### Analysis of the structure and function of the worm AH domain

A region of the alpha helical domain of mouse complexin binds tightly to the assembled SNARE bundle (*Figure 1A*), positioned in the groove formed by synaptobrevin and syntaxin (*Bracher et al., 2002*; *Chen et al., 2002*). This so-called 'central helix' (CH) is deeply conserved across phylogeny (76% identity between mammals and nematodes), whereas the adjacent helical sequence corresponding to the mouse accessory helix (AH) is much more heterogeneous, with only 20% identity between mammals and nematodes based on the 18 residues N-terminal to CH (*Figure 1B*). To compare AH function in mouse and worm, we first established whether the worm complexin protein CPX-1 possesses a stable helical

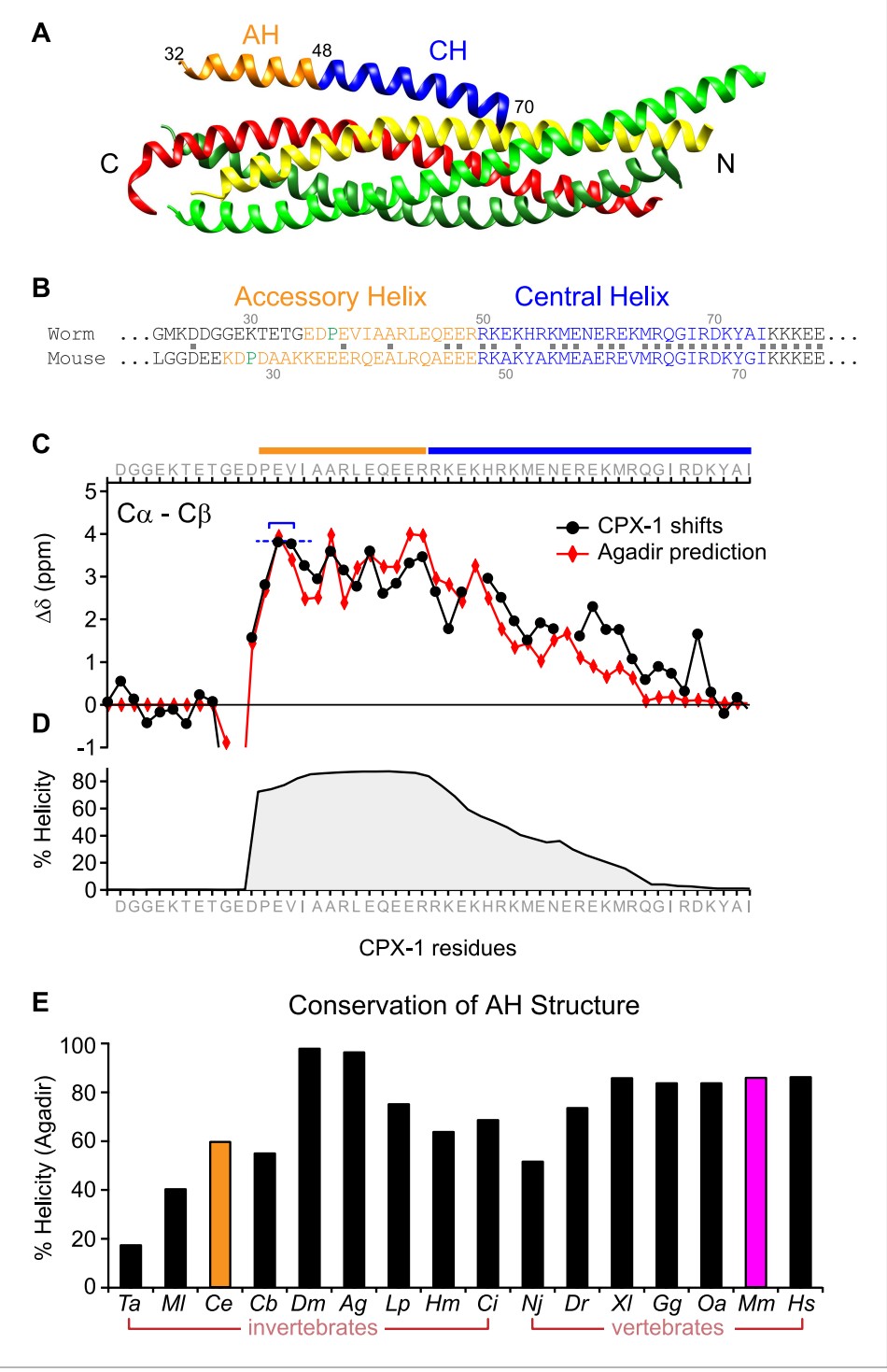

**Figure 1**. The worm AH domain forms a stable helix and this structure is deeply conserved across phylogeny. (**A**) Ribbon diagram of the mammalian complexin-SNARE crystal structure (*Chen et al., 2002*) using PDB code 1KIL. Cytoplasmic SNARE domains from synaptobrevin (*red*), Syntaxin (*yellow*), and SNAP-25 (*green*). Mouse Cpx1 (residues 26–83) is divided into the accessory helix (AH–*orange*) and central helix (CH–*blue*). (**B**) Sequence alignment of the accessory helix (*orange*) and central helix (*blue*) for *C. elegans* CPX-1 (*worm*) and *M. musculus* Cpx1 (*mouse*). Amino acid identity indicated with gray squares in between the sequences. Helix-breaking prolines indicated in green. (**C**) Cα-Cβ shifts from a truncated worm CPX-1 peptide missing the C-terminal domain (residues 1–77, *black circles*) are compared with Agadir predictions for the Cα shifts (*red diamonds*). The Agadir predicted Cα shift
*Figure 1. Continued on next page*

*Figure 1. Continued*

values at residues E38 and V39 were normalized to the experimentally determined Cα-Cβ shift values at those sites (blue bracket), allowing for a comparison of AH and CH shift predictions. (**D**) Predicted helical state of each residue (Agadir) is shown for CPX-1. (**E**) Summary of Agadir helix prediction for the AH domain (defined by the average helicity of 18 residues N-terminal to the CH domain) across 16 species: *Trichoplax adhaerens Ta, Mnemiopsis leidyi Ml, Caenorhabditis elegans Ce (orange), Caenorhabditis briggsae Cb, Drosophila melanogaster Dm, Anopheles gambiae Ag, Loligo pealei Lp, Hirudo medicinalis Hm, Narke japonica Nj, Ciona intestinalis Ci, Danio rerio Dr, Xenopus laevis Xl, Gallus gallus Gg, Ornithorhynchus anatinus Oa, Mus musculus Mm (pink), Homo sapiens Hs.*
The following figure supplements are available for figure 1:
**Figure supplement 1**. Alpha helical regions of complexin across phylogeny.
**Figure supplement 2**. Evolutionary conservation of helicity across >0.5 billion years.

region adjacent to its CH analogous to mouse complexin (*Pabst et al., 2000*; *Bracher et al., 2002*; *Chen et al., 2002*). Computational predictions based on amino acid sequence indicated a highly stable helical region including 30 residues between P37 and G66 encompassing the AH and half of the CH domain (*Figure 1C,D*) (*Munoz and Serrano, 1997*). This domain was confirmed to be helical by solution-state NMR spectroscopy on recombinant full-length worm CPX-1 (*Snead et al., 2014*) as well as on a truncated version lacking its C-terminal domain (*Figure 1C,D*), validating the computational predictions for this protein. The stable helical structure of the AH domain is predicted to be deeply conserved across phylogeny based on the analysis of complexin sequences in 16 diverse metazoan species from seven phyla ranging from *Trychoplax* to human (*Figure 1—figure supplements 1–2*). Thus both mouse and worm complexin possess a stable alpha helical domain N-terminal to the CH despite sharing little sequence homology, and this is likely to be a universal feature of complexin.

To investigate the functional significance of the AH domain in worm complexin, two deletion variants of CPX-1 missing either 12 (ΔAH$_{short}$) or 20 (ΔAH$_{long}$) residues of the AH were expressed in *cpx-1* mutants (*Figure 2A*). All transgenic rescue experiments reported in this study utilized a functional CPX-1::GFP fusion protein, and transgene expression was monitored by imaging neuromuscular junction (NMJ) fluorescence in living intact animals (*Martin et al., 2011*; *Wragg et al., 2013*) (*Figure 2—figure supplement 1*). Loss of CPX-1 caused a 12-fold increase in the rate of spontaneous fusion in the absence of external calcium at cholinergic NMJs (*Hobson et al., 2011*; *Martin et al., 2011*; *Wragg et al., 2013*) (*Figure 2B,C*). Rescue with full-length CPX-1 completely restored the basal synaptic vesicle (SV) fusion rate (3.3 ± 0.9 Hz for wild type, 2.7 ± 0.5 Hz for FL rescue) whereas rescue with the ΔAH$_{short}$ variant only partially reversed the increased rate of spontaneous fusion (39.8 ± 6.2 Hz for *cpx-1*, 20.8 ± 5.6 Hz for ΔAH$_{short}$ rescue). Thus CPX-1 retained a limited ability to inhibit fusion in the absence of the AH domain. In all cases, the muscle miniature EPSC amplitude was unaffected (*Figure 2D*), consistent with the neuronal expression of CPX-1 in *C. elegans* (*Martin et al., 2011*). *cpx-1* mutants exposed to the cholinesterase inhibitor aldicarb paralyzed more rapidly than wild-type animals (*Figure 2E,F*) as described previously (*Hobson et al., 2011*; *Martin et al., 2011*; *Wragg et al., 2013*). Consistent with the NMJ recordings, rescue with ΔAH$_{short}$ or ΔAH$_{long}$ only partially restored wild-type sensitivity whereas full-length CPX-1 completely rescued aldicarb sensitivity (*Figure 2F*). The helix-breaking proline in position 37 is shared by all of the published nematode genomes, so the short AH domain is a common feature of this phylum. However, deletion of proline 37 does not impair CPX-1 inhibitory function, indicating that the short AH domain is not an essential feature of nematode complexin structure (data not shown). Taken together, these structural and functional results demonstrate that, despite the lack of sequence conservation between worm and mouse AH, the worm AH domain comprises a highly stable alpha helix that plays a major role in CPX-mediated inhibition of spontaneous SV fusion.

## Conservation of AH domain mechanism between worm and mouse

The similar structures of worm and mouse complexin AH domains suggest that AH function could be conserved between these highly divergent species. Indeed, several studies have indicated that mouse and fly AH domains contribute to an inhibitory activity of complexin (*Xue et al., 2007*; *Giraudo et al., 2008*;

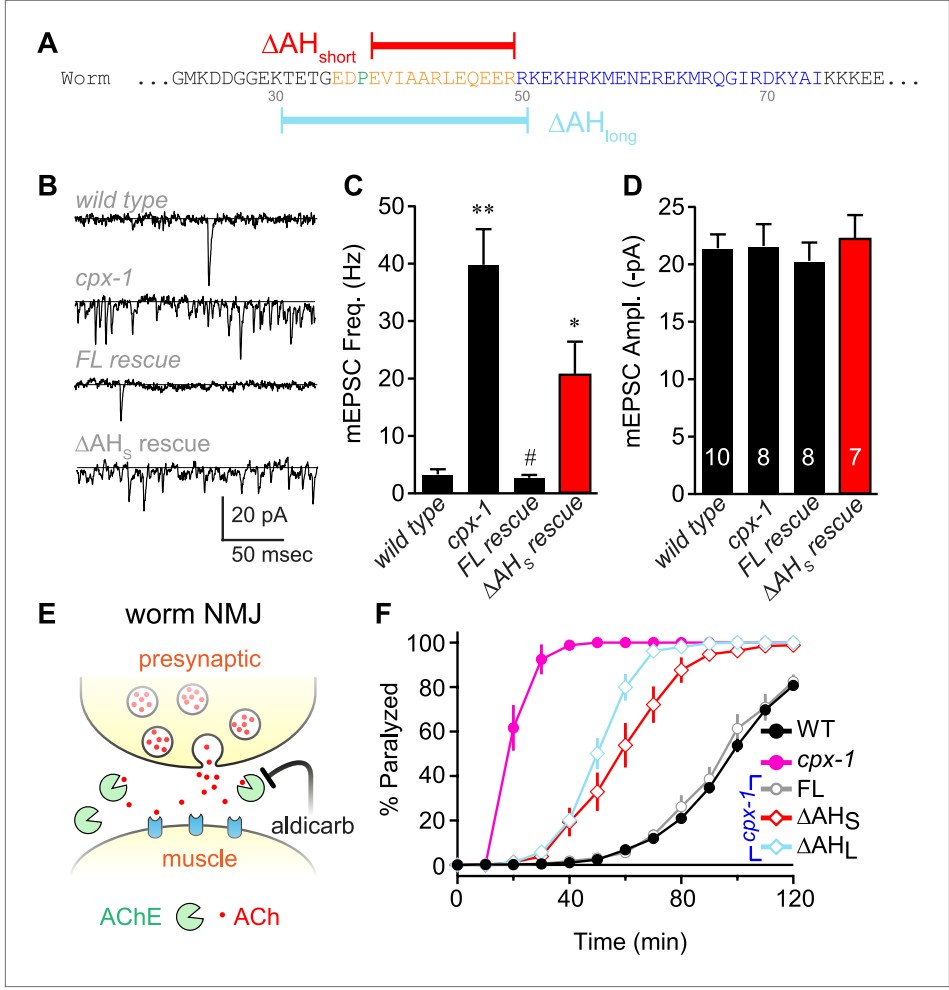

**Figure 2**. The worm AH contributes to CPX-1 inhibition of spontaneous vesicle fusion. (**A**) Two deletions within the worm AH domain were used: *ΔAH*ₛₕₒᵣₜ (35–49, *red*) and *ΔAH*ₗₒₙg (30–50, *aqua*). (**B**) Examples of spontaneous EPSCs in zero external Ca²⁺ for wild-type, *cpx-1*, and transgenic animals expressing full-length CPX-1 (*FL rescue*) and the short AH deletion (*ΔAHₛₕₒᵣₜ rescue*). Average spontaneous EPSC Rate (**C**) and EPSC amplitude (**D**) for the genotypes indicated in **B**. Data are mean ± SEM and the number of independent assays is indicated for each genotype. Using Tukey–Kramer statistics for multiple comparisons, ** denotes significantly different from wild type, # significantly different from *cpx-1* but not wild type, * significantly different from both wild type and *cpx-1* (p < 0.01). (**E**) Cartoon of aldicarb acting at the worm cholinergic neuromuscular junction. Acetylcholine (*ACh, red*) is hydrolyzed by cleft cholinesterases (*AChE, green*). Aldicarb inhibits AChE causing an elevation in ACh and eventual paralysis depending on the level of exocytosis. (**F**) Paralysis time course on 1 mM aldicarb for wild-type (*black filled circles, n = 36*), *cpx-1* (*pink filled circles, n = 10*), full-length rescue CPX-GFP (*gray open circles, n = 10*), *ΔAH*ₛₕₒᵣₜ rescue (*red open diamonds, n = 10*), and *ΔAH*ₗₒₙg rescue (*aqua open diamonds, n = 10*).

The following figure supplement is available for figure 2:

**Figure supplement 1**. Axonal protein abundance for CPX-1 transgenes.

*Chicka and Chapman, 2009*; *Giraudo et al., 2009*; *Kummel et al., 2011*; *Cho et al., 2014*; *Lai et al., 2014*; *Trimbuch et al., 2014*). However, the AH may operate through distinct mechanisms in distantly related species. To test for conservation of mechanism, the endogenous worm AH was replaced with the mouse AH and expressed in *cpx-1* mutants. Since the length of the AH is not identical between worm and mouse, two versions of the chimeric complexin were generated: a longer version encompassing mouse residues 24–47 replaced the worm residues 26–49 (mouse AH), and a shorter AH swap replaced 38–49 with mouse residues 36–47 (mouse AHₛₕₒᵣₜ) (*Figure 3A*). Recordings from the NMJs of *cpx-1* mutants expressing the mouse AH demonstrated that CPX-1 is fully functional with a

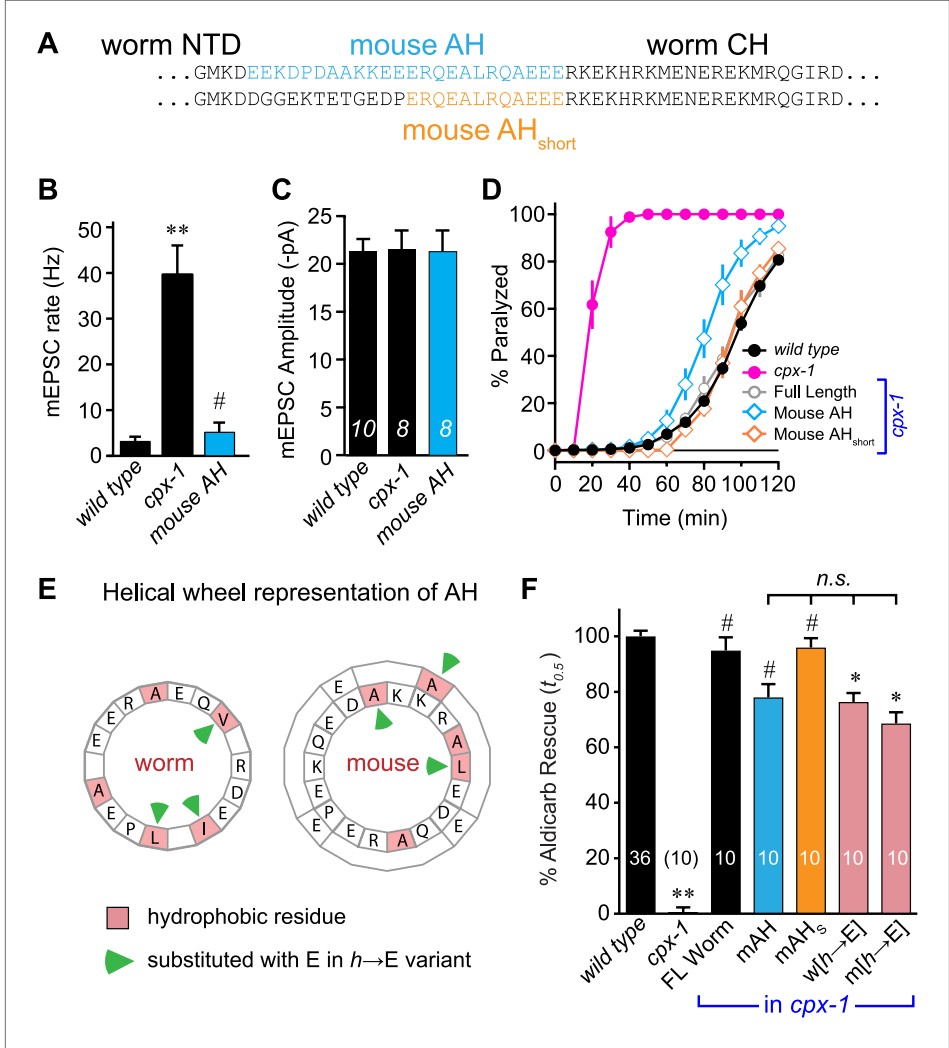

**Figure 3**. Mouse AH functions in worm CPX-1 and neither domain requires hydrophobic residues. (**A**) Two chimeric CPX-1 constructs substituting worm AH with the mouse AH. The long form (mouse AH) substitutes worm residues 26–49 with mouse residues 24–47 whereas the short form (mouse AH$_{short}$) substitutes worm residues 38–49 with mouse residues 36–47. Average spontaneous mEPSC rates (**B**) and amplitudes (**C**) for wild-type, *cpx-1*, and transgenic rescue of *cpx-1* expressing the mouse AH chimera (*mouse AH, blue*). (**D**) Average paralysis time course in 1 mM aldicarb for wild-type (*black filled circles*), *cpx-1* (*pink filled circles*), full-length rescue (*gray open circles*), mouse AH rescue (*blue open diamonds*), and mouse AH$_{short}$ rescue (*orange open diamonds*). (**E**) Worm and mouse AH domain residues shown on a helical wheel diagram with hydrophobic residues indicated in red and hydrophobic substitutions with glutamate indicated with green arrow heads. (**F**) Percent rescue of wild-type paralysis kinetics using the time to 50% paralysis ($t_{0.5}$) for the five genotypes shown in **D** as well as a worm CPX-1 variant with V39E, I40E, L44E substitutions (w[$h{\rightarrow}$E]) and a mouse AH$_{short}$ variant with A40E, L41E, A44E substitutions (m[$h{\rightarrow}$E]). Note that the residue positions are labeled based on their location in CPX-1 and mCpx1 respectively. Data are mean ± SEM with sample sizes as indicated on the bar graphs. Using Tukey–Kramer statistics for multiple comparisons, ** denotes significantly different from wild type, # significantly different from *cpx-1* but not wild type, * significantly different from both wild type and *cpx-1* ($p < 0.01$), *n.s.* is not significant.

The following figure supplements are available for figure 3:

**Figure supplement 1**. Conservation of hydrophobic moments of complexin.

**Figure supplement 2**. Conservation of charge density in the AH domain.

mouse AH domain (*Figure 3B,C*). Both mouse AH and mouse AH$_{short}$ restored near wild-type aldicarb sensitivity, further indicating that the mouse AH is functional in the context of worm CPX-1 (*Figure 3D*). The precise length of the AH is not conserved across species (*Figure 1—figure supplement 1*), and the spacing between the N-terminal domain and the CH is not likely to play a critical role in CPX-1 function since the NTD can be deleted without significantly impairing CPX-1 inhibitory function in worm (*Hobson et al., 2011*; *Martin et al., 2011*).

We next examined the phylogenetic conservation among several AH domains assuming that shared features are more likely to indicate conservation of AH domain function. Comparing AH domain properties in 16 species across seven phyla, three conserved features were apparent: a region of hydrophobic residues along the helix (*Figure 3—figure supplement 1*), stable helicity (*Figure 1E*), and high negative charge density (*Figure 3—figure supplement 2*). One recent model of AH function proposes that an interaction between the hydrophobic AH residues and the tSNARE complex prevents full SNARE assembly thereby inhibiting SV fusion (*Krishnakumar et al., 2011*; *Kummel et al., 2011*). To test this model in *C. elegans*, several hydrophobic residues in the AH domain (*Figure 3E*) were replaced with the charged residue, glutamate (h→E) and this CPX-1 variant was expressed in *cpx-1* mutants. As shown in *Figure 3F*, the AH(h→E) was fully functional in the absence of all hydrophobic side chains. Moreover, the short mouse AH chimera was also functional even when its hydrophobic residues were replaced with glutamates. Similar observations have been reported at the fly NMJ using mouse Cpx1 (*Cho et al., 2014*). These findings indicate that hydrophobic residues in the AH domain do not play a critical conserved role in complexin function.

## Impact of secondary structure on AH domain function

Previous studies on mouse complexin suggested that the helical structure of the AH domain is important for stabilizing the CH domain (*Pabst et al., 2000*; *Chen et al., 2002*) and for the inhibitory function of mCpx1 (*Xue et al., 2007*), but the reason for this requirement remains unclear. While several potential roles for the AH in mediating protein–protein interactions have been proposed, the AH may simply serve to nucleate and propagate helical structure into the CH region (*Chen et al., 2002*), but this idea has never been tested. To explore this possibility, a helix-breaking proline was inserted into the AH domain (R43P), and the effects on AH domain secondary structure of a recombinant truncated form of the mutant protein missing its C-terminal domain (ΔCT) were examined by NMR spectroscopy (*Figure 4A*). Furthermore, because conversion from random coil to alpha helix is a highly cooperative process, helicity in the CH domain is also predicted to decrease for the R43P mutant (*Munoz and Serrano, 1997*). Indeed, decreased NMR carbon secondary shifts were observed throughout the AH and extending well into the CH domain, confirming decreases in both nucleation and propagation of the helical conformation (*Figure 4B*). Circular dichroism (CD) spectroscopy provides another measure of overall alpha helical structure (*Greenfield and Fasman, 1969*; *Saxena and Wetlaufer, 1971*; *Rohl and Baldwin, 1997*). Absorption at 222 nm was monitored in recombinant ΔCT protein while titrating in 2,2,2-trifluoroethanol (TFE), a co-solvent known to stabilize alpha helices in solution (*Nelson and Kallenbach, 1986*; *Segawa et al., 1991*; *Shiraki et al., 1995*). The increase in alpha helical structure with increasing concentration of TFE can be used to measure the stability and cooperativity of alpha helix formation. While some of the cooperativity arises from coordination of multiple TFE molecules (*Berkessel et al., 2006*), intramolecular propagation of helical structure will also contribute. As shown in *Figure 4C*, the R43P variant displayed a lower propensity for helix formation with a lower cooperativity, consistent with both decreased helix nucleation and propagation, as also evident from both the computational predictions and the NMR data. In living worms, inserting a proline into the AH domain completely eliminated AH function since the CPX(R43P) rescue was indistinguishable from the ΔAH$_{short}$ rescue in both electrophysiological (*Figure 4D,E*) and behavioral (*Figure 4F*) assays of synaptic function. Thus, inhibition of spontaneous fusion requires a helical AH domain in vivo.

## Tests of the helix nucleation/propagation hypothesis

These results provide support for a model in which nucleation of an alpha helix within the AH domain and propagation of this helix into the CH domain is required for CPX inhibitory function. Three predictions arise from this hypothesis: First, there should be a direct correlation between helical stability and inhibitory function. Second, disruption of helix propagation into the CH domain should impair CPX inhibition. And third, replacement of the AH with a non-native alpha helical domain should functionally substitute for the endogenous AH sequence and stabilize the CH domain.

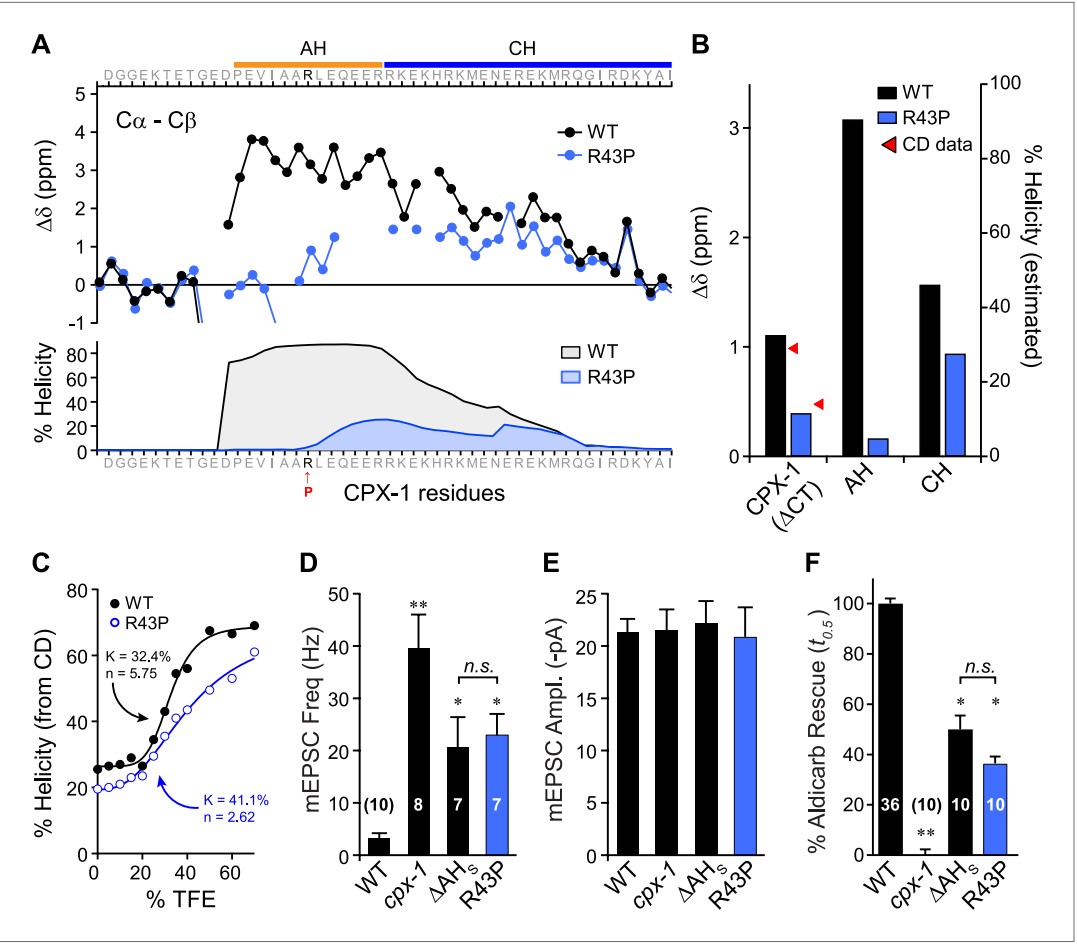

**Figure 4**. Disrupting AH helix stability impairs CPX-1 inhibitory function. (**A**) NMR derived Cα-Cβ shifts from either wild-type (*black*) or R43P (*blue*) worm CPX-1 peptide missing the C-terminal domain. R43P is indicated in red. Below, the predicted helical content using Agadir for wild-type (*black*), and R43P complexin (*blue*). (**B**) Average secondary chemical shift for wild-type (black) and R43P (blue) complexin either over the entire peptide (residues 1–77, left), AH domain (37–49, middle), or CH domain (50–74, right). The average helical content was estimated by dividing the chemical shift by 3.4 (average shift of 100% helical peptide). The helical content was also measured by CD spectroscopy (red arrowheads). (**C**) Helical content for wild type (*black*) and R43P (*blue*) complexin was measured by CD spectroscopy for increasing concentrations of 2,2,2-trifluoroethanol (TFE). The resulting dose–response data was fit to a simple equilibrium binding curve with equilibrium constants and Hill coefficients indicated on the graph. Average spontaneous EPSC Rate (**D**) and EPSC amplitude (**E**) for *wild-type*, *cpx-1*, and either the ΔAH$_{short}$ or R43P rescuing transgene expressed in *cpx-1* as indicated. (**F**) Sensitivity to aldicarb was quantified by the average time to 50% paralysis and then normalized to wild-type and *cpx-1* mutant animals. On this scale, rescue with ΔAH$_{short}$ or R43P variants of CPX-1 partially restored wild-type aldicarb sensitivity. Data are mean ± SEM and the number of independent assays is indicated for each genotype. Using Tukey–Kramer statistics for multiple comparisons, ** denotes significantly different from wild type, * significantly different from both wild type and *cpx-1* (p < 0.01), *n.s.* is not significant.

The first prediction was examined by creating a series of CPX variants predicted computationally to feature decreasing helical stability (***Figure 5A,B***). Helical stability dropped monotonically as the residue at position 43 changed from R → V → F → P. Notably, the predictions also indicate a parallel decrease in the helicity of the CH helix, consistent with the nucleation/propagation hypothesis. This AH series was expressed in *cpx-1* mutants, and rescue was quantified using aldicarb sensitivity. Supporting this hypothesis, transgenic animals expressing this series of CPX-1 variants display a level of aldicarb sensitivity that mirrors the helical stability (***Figure 5C,D***). To test the second prediction, helical propagation was blocked by inserting a Gly–Gly (GG) between the AH and CH domains. Because glycines destabilize helical structure, GG insertions are expected to disrupt helical propagation (***O'Neil and***

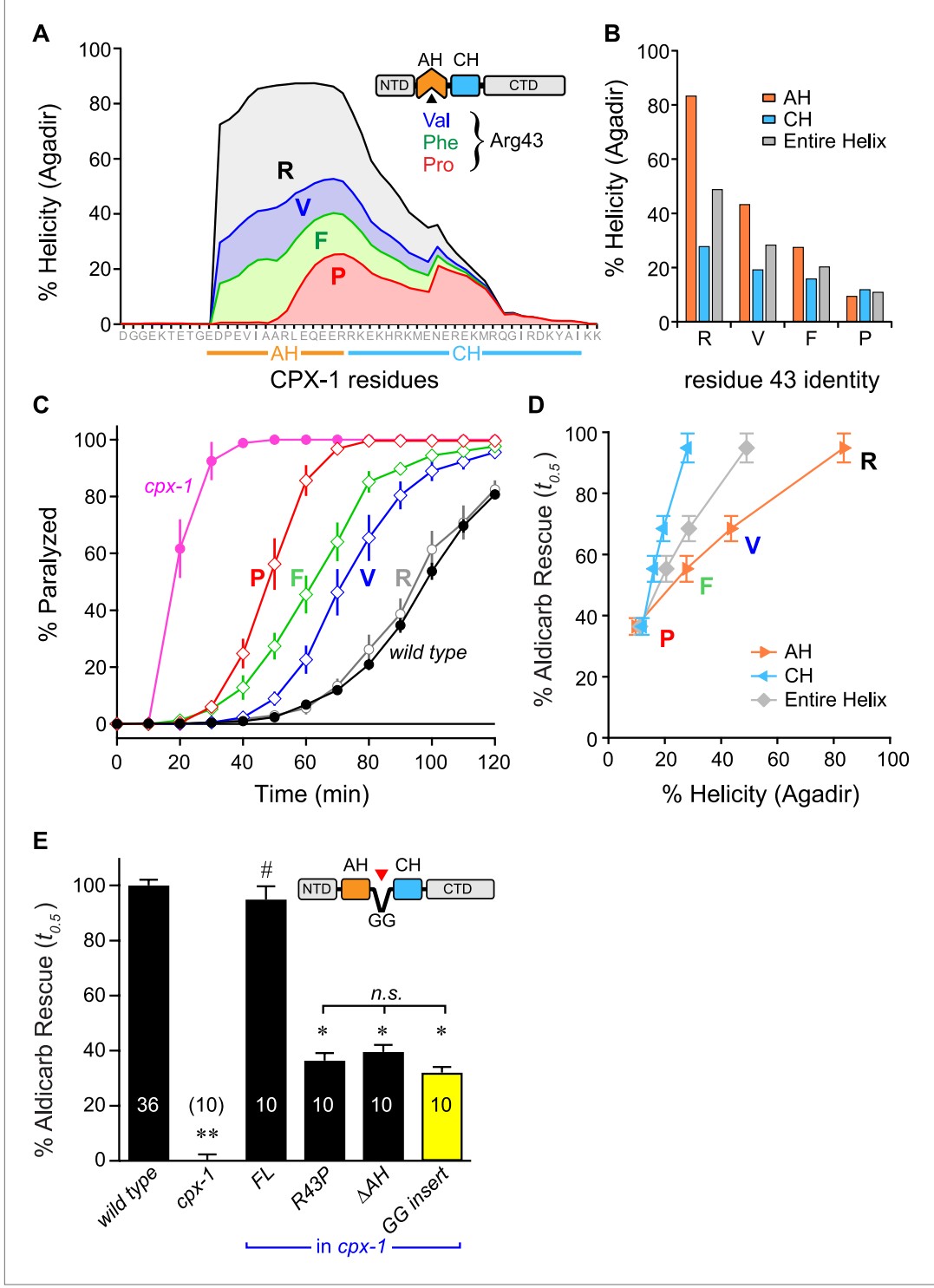

**Figure 5**. Stability of the AH and its propagation into the CH domain are required for CPX-1 inhibitory function. (**A**) AH and CH helical content based on Agadir predictions is plotted for wild-type CPX-1 and three variants with a single substitution at residue 43 as indicated. (**B**) Average helical content of the AH domain (*orange*), CH domain (*blue*), or entire helical region (*gray*) for four residues in position 43: Arg, Val, Phe, and Pro. (**C**) Average paralysis time course on one millimolar aldicarb for wild-type (black filled circles) and *cpx-1* animals (pink filled circles), as well as four rescuing CPX-1 transgenes: wild type CPX-1 (R, gray open circles), R43V (V, blue diamonds), R43F (F, green diamonds), and R43P (P, red diamonds). (**D**) Normalized rescue of the $t_{0.5}$ for paralysis for each of the four transgenic
*Figure 5. Continued on next page*

*Figure 5. Continued*

rescue strains is plotted vs the predicted helicity for either the AH domain (*orange*), the CH domain (*blue*), or the entire helical region (*gray*). (**E**) Normalized rescue of aldicarb sensitivity ($t_{0.5}$) for wild-type and *cpx-1* animals as well as three transgenic rescue strains: full-length wild-type CPX-1 (*FL*), R43P, the long AH deletion (*ΔAH*), and the GG insert in between the AH and CH domains (*GG insert*). Data are mean ± SEM with *n* = 10 experiments for all strains except wild type (*n* = 36). **p < 0.01 different from wild-type, # is p < 0.01 different from *cpx-1* but not wild type. *p < 0.01 different from wild type and *cpx-1* animals. Significance was determined by Tukey–Kramer method.

*DeGrado, 1990*; *Blaber et al., 1993*) as supported by the computational predictions of AH and CH helical content. Rescue of *cpx-1* mutants with the GG variant demonstrated that insertion of a GG between the AH and CH was equivalent to deleting the entire AH domain (*Figure 5E*). This disruption is not due to a shift in the orientation of AH domain hydrophobic residues (a consequence of inserting two residues) since inserting three glycines produced a similar loss of function (data not shown). Thus helix propagation from the AH domain to the CH domain appears to play a major role in CPX-1 inhibitory function, in agreement with a similar finding at the fly NMJ (*Cho et al., 2014*).

As a third test of the helix nucleation/propagation hypothesis, the endogenous AH domain was replaced with an artificial alpha helix based on a Glu-Ala-Ala-Lys (EAAK) motif repeated seven times (*Figure 6A*). This design was predicted to form a stable alpha-helical structure in solution (*Figure 6B*) (*Marqusee and Baldwin, 1987*). CD spectroscopy of TFE titrations on recombinant CPX-1(ΔCT) containing the 7-turn helix revealed that this variant is more helical than wild type (*Figure 6C*). The increased helicity values could arise because 24 residues of the CPX-1 AH were replaced with 30 residues of helical EAAK repeat, so a larger fraction of the whole polypeptide will necessarily be helical. As an independent measure of helical stability, we compared the helix-coil equilibrium constants derived from TFE titrations and found that the 7-turn polypeptide forms a more stable helix than wild-type CPX-1 (17.9% for 7-turn vs 32.4% for wild type). Thus, the 7-turn artificial alpha helix provides a stable helical substitute for the endogenous AH domain irrespective of its increased length. Additionally, the 7-turn construct displayed a lower apparent cooperativity than wild-type CPX-1 (2.8 for 7-turn vs 5.6 for wild type), but the true cooperativity of the 7-turn polypeptide was underestimated since it was quite helical even at 0% TFE (*Figure 6C*). Surprisingly, the 7-turn helix fully rescued both aldicarb sensitivity and suppression of spontaneous fusion at cholinergic NMJs (*Figure 6D–F*). Moreover, the AH(7-turn) sequence is electrostatically neutral overall, yet it fully restored CPX inhibition. Therefore, the conserved negative charge density of the AH domain was not essential for CPX inhibition. In fact, the charge density of the rescuing transgenes used in this study generally did not correlate with their function in vivo (*Figure 6—figure supplement 1*). The EAAK motif also creates a strong hydrophobic moment, but the orientation of the moment is rotated approximately 90° relative to wild-type CPX-1, indicating that AH function is not sensitive to the positioning of the hydrophobic residues (*Figure 6—figure supplement 2*). Together with the progressive AH destabilizing substitutions and GG insertion, these results reveal that a critical feature of AH function is the nucleation and invasion of a stable alpha helix into the CH domain.

## Discussion

Complexin is a potent regulator of neurotransmitter release and studies from several synaptic preparations have revealed distinct roles for complexin including both stimulatory and inhibitory functions (*Huntwork and Littleton, 2007*; *Xue et al., 2007*; *Chicka and Chapman, 2009*; *Malsam et al., 2009*; *Maximov et al., 2009*; *Hobson et al., 2011*; *Martin et al., 2011*; *Kaeser-Woo et al., 2012*; *Wragg et al., 2013*). It is not clear if these functions are always present in different species and whether they share the same underlying mechanisms. The results of this study indicate that although the AH domain is poorly conserved at the primary sequence level, at least some of its underlying inhibitory function and mechanism is preserved between species. Disruption of helix nucleation and propagation from the AH to the CH prevents CPX-mediated inhibition of spontaneous vesicle fusion. Furthermore, CPX-1 retains some inhibitory function in the absence of the AH but not in the absence of the CH (*Martin et al., 2011*), consistent with the idea that inhibition derives from the CH. While the primary sequence of the CH is evolutionarily conserved in parallel with its SNARE binding partners, only the helical stability of the AH is retained across phylogeny. Thus, the various proposals of AH protein–protein interactions and electrostatic repulsion effects are not fundamental requirements of CPX inhibitory function at the synapse.

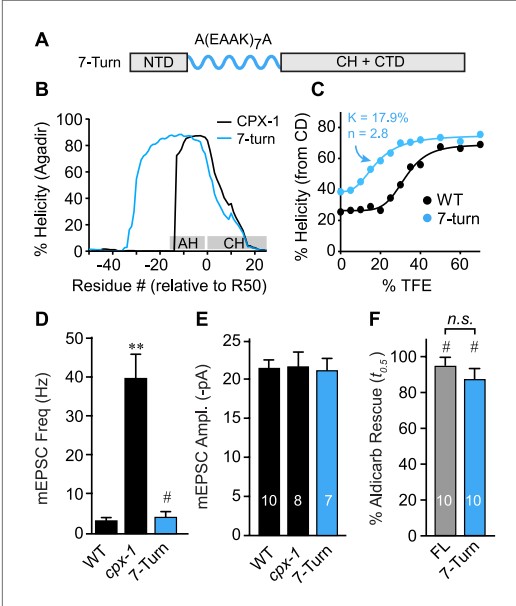

**Figure 6**. The AH domain can be functionally replaced by a non-native helix. (**A**) Schematic of the helix substitution strategy. A(EAAK)$_7$A sequences were substituted for residues 26–49 in the worm AH domain. (**B**) Agadir prediction for helical stability of the 7-turn helix motif compared to wild-type AH. (**C**) Helical content for the 7-turn construct was measured by CD spectroscopy, and the TFE dose–response data were fit as in *Figure 4*. Average spontaneous EPSC Rate (**D**) and EPSC amplitude (**E**) for *wild-type*, *cpx-1*, and the 7-turn rescuing transgene expressed in *cpx-1* as indicated. (**F**) Sensitivity to aldicarb was quantified by monitoring the average time to 50% paralysis normalized to wild-type and *cpx-1* mutant animals, and plotted for full-length wild-type CPX-1 (FL), 5-turn substitution (5 Turn), and 7-turn substitution variants expressed in *cpx-1* mutants. Data are mean ± SEM and the number of independent assays is indicated for each genotype. Using Tukey–Kramer statistics for multiple comparisons, ** denotes significantly different from wild type, # significantly different from *cpx-1* but not wild type, * significantly different from both wild type and *cpx-1* (p < 0.01), *n.s.* is not significant.

The following figure supplements are available for figure 6:

**Figure supplement 1**. No correlation between charge density and function.

**Figure supplement 2**. Hydrophobic moments of functional and nonfunctional CPX transgenes.

## Evolution of complexin

Why compare complexins from multiple divergent species? Several features of the AH domain appear to be conserved across more than half a billion years of evolution based on the species examined here, so an evolutionary comparison provides insight into critical aspects of the underlying mechanisms. Beyond their primary sequence, the two most striking conserved features of the AH domain are its high negative charge density and the distribution and orientation of its hydrophobic residues (*Figure 3—figure supplements 1–2*). Although the deep conservation of charge density and hydrophobic moment suggests that they play a role in complexin function, the inhibition of spontaneous fusion does not require either feature in worm (this study) and hydrophobicity is superfluous in fly as well (*Cho et al., 2014*). Perhaps other roles of complexin utilize these properties. In contrast to charge and hydrophobicity, a stable helix is both deeply conserved and required for complexin inhibitory function. How conserved is helicity vs primary protein sequence? The SNARE-binding central helix is the defining motif of complexin homologs based on primary protein sequences (*Pabst et al., 2000*; *Brose, 2008*). Two of the most distantly related complexin genes reported belong to the placozoan *Trichoplax adherens* (*Ta*) and the ctenophore *Mnemiopsis leidyi* (*Ml*) (*Flicek et al., 2014*). Interestingly, the CH domains of these representatives of basal animal phyla, share 44% identity with the human Cpx1 CH. However, whereas the AH domain of *Ta* shares 40% identity with the human AH domain, there is only 10% sequence conservation between *Ml* and human AH domains. The predicted AH and CH structures for human and *Ml* both contain extended regions of stable helix propagation similar to worm, whereas *Ta* is predicted to have only a modest degree of stable helical structure on its own (*Figure 1—figure supplement 2*). This provides an evolutionary example of how primary sequence and helicity do not necessarily change in parallel in the AH domain. A speculative explanation for the noticeable increase in AH domain helical stability progressing from ctenophore (36%) to worm (54%) to mammal (>80% in human) is the higher body temperatures of warm-blooded animals compared to soil- and marine invertebrates. Helical stability is highly dependent on ambient temperature, so the higher body temperatures of mammals and many other vertebrates may necessitate more stable helical sequences (*Privalov, 1982*; *Scholtz and Baldwin, 1992*).

## Models of AH mechanism

Inhibition by complexin depends on the integrity of its AH domain in all studies where this domain has been examined (*Xue et al., 2007*; *Giraudo et al., 2008*; *Chicka and Chapman, 2009*; *Giraudo et al., 2009*;

*Xue et al., 2009*; *Cho et al., 2010*; *Yang et al., 2010*; *Krishnakumar et al., 2011*; *Kummel et al., 2011*; *Martin et al., 2011*; *Cho et al., 2014*). Several models for AH-domain function have been proposed over the past few years. Based on the unusual stability of the helical structure of the AH motif noted in early studies of the complexin-SNARE complex, the AH has been speculated to nucleate and propagate helical structure into the SNARE-binding CH (*Chen et al., 2002*). Several groups have proposed that the AH domain of mouse Cpx1 competes with the C-terminal region of VAMP2 for a binding site on the tSNAREs thereby preventing full SNARE assembly (*Giraudo et al., 2009*; *Lu et al., 2010*). The AH is also thought to mediate an intermolecular interaction between neighboring SNARE complexes, and this *trans* Cpx/SNARE array was proposed to underlie complexin-mediated inhibition based on in vitro assays of membrane fusion (*Krishnakumar et al., 2011*; *Kummel et al., 2011*). A recent study examining the *trans* Cpx/SNARE array model in the fly NMJ found that mutations expected to disrupt the hydrophobic interaction between the AH domain and the tSNAREs did not impair inhibition of spontaneous SV fusion in vivo (*Cho et al., 2014*) in agreement with the hydrophobic residue substitutions used in the present study. Based on molecular dynamics simulations, another model conjectures that the AH domain can form a tight complex with the SNARE bundle and can also stabilize a partially-assembled state by binding directly to the C-terminal region of VAMP2 (*Bykhovskaia et al., 2013*). In contrast to these AH-SNARE interaction models, electrostatic repulsion between the AH domain and membranes has been suggested to be inhibitory (*Trimbuch et al., 2014*). Thus a broad range of models and features have been put forward to describe AH domain function.

### How does the AH domain function?

Our results suggest that the mechanism of AH function is deeply conserved and relatively independent of the primary protein sequence, hydrophobicity, length, or net charge density, inconsistent with models that rely on specific AH-protein interactions or electrostatic AH-membrane interactions. However, the mechanistic effect of stabilizing the CH domain remains unclear. An appealing model is that a stabilized CH alpha helix interacts more efficiently with the assembling SNARE bundle thereby promoting the SNARE interaction required for complexin inhibition. Indeed, CH binding to SNAREs is reduced in the absence of the AH in mouse Cpx1 (*Xue et al., 2007*). Alternatively, the CH domain could interact with an as yet unidentified synaptic protein, and this binding would then require stabilization by the AH. The models based on hydrophobic AH interactions or electrostatic contributions are not supported by the experiments described here, but these mechanisms may be more prominent in other species. Alternatively, the high negative charge and hydrophobic moment of the AH domain may be relevant for other complexin functions. Nevertheless, our data indicate that the nucleation and propagation function of the AH domain is universal for proper complexin function across species. Further exploration of the CH and its binding partners is required for a detailed mechanistic understanding of complexin inhibitory action at the synapse.

## Materials and methods

### Strains

Animals were maintained at 20°C on agar nematode growth media seeded with OP50 bacteria as previously described (*Brenner, 1974*). Strains employed in this study include: N2 Bristol and *cpx-1* (*ok1552*).

#### Transgenic strains

JSD0291: *tauIS90;cpx-1* [P*snb-1::*CPX-GFP].
JSD0462: *tauIS125;cpx-1*[P*snb-1::*CPX (mouse AH)-GFP].

#### Extrachromosomal arrays

JSD0472: *tauEx105;cpx-1* [P*snb-1::*CPX (worm h→E)-GFP].
JSD0651: *tauEx189;cpx-1*[P*snb-1::*CPX(R43P)-GFP].
JSD0654: *tauEx192;cpx-1*[P*snb-1::*CPX(mouse AH$_{short}$)-GFP].
JSD0655: *tauEx193;cpx-1*[P*snb-1::*CPX(ΔAH$_{short}$)-GFP].
JSD0657: *tauEx195;cpx-1*[P*snb-1::*CPX(ΔAH$_{long}$)-GFP].
JSD0714: *tauEx232;cpx-1*[P*snb-1::*CPX(mouse h→E)-GFP].
JSD0775:*tauEx265;cpx-1* [P*snb-1::*CPX(AH-GG-CH)-GFP].
JSD0778: *tauEx273;cpx-1* [P*snb-1::*CPX(R43F)-GFP].

JSD0777: *tauEx272;cpx-1* [P*snb-1*::CPX(R43V)-GFP].
JSD0780: *tauEx288;cpx-1* [P*snb-1*::CPX(7 turn helix)-GFP].

All extrachromosomal arrays were expressed in *cpx-1(ok1552)* null mutant animals and protein expression in the NMJ was quantified as described previously (*Wragg et al., 2013*; *Snead et al., 2014*). All strains used in this study fell within previously published levels of expression that can fully rescue the *cpx-1* mutant phenotype (*Figure 2—figure supplement 1*). See 'Imaging' section for methods.

## Acute aldicarb sensitivity

To measure aldicarb sensitivity, 20–30 young adult animals were placed on agar plates containing 1 mM aldicarb (Watson International, China). Worms were scored for paralysis at ten minute intervals for 2 hr. Each genotype was coded, tested 10 times blindly, and the paralysis curves were generated by averaging paralysis time courses for each plate as described previously (*Dittman and Kaplan, 2008*). Percent rescue based on $t_{0.5}$ was calculated by first interpolating the time at which 50% of the worms paralyzed for each trial, averaging the single-trial $t_{0.5}$ values together, and then normalizing to 100% rescue for wild-type $t_{0.5}$ and 0% rescue for *cpx-1* $t_{0.5}$ values according to:

$$\%R_{strain} = 100 \cdot \frac{t_{0.5}\left[strain\right] - t_{0.5}\left[cpx\right]}{t_{0.5}\left[WT\right]}$$

## Imaging

To control for protein expression levels in the extrachromosomal arrays, animals were first immobilized using 2,3-butanedione monoxime (30 mg/ml, Alfa Aesar, Ward Hill, MA), mounted on 2% agarose pads, in M9 buffer (22.0 mM $KH_2PO_4$, 42.3 mM $Na_2HPO_4$, 85.6 mM NaCl, and 1.0 mM $MgSO_4$), and imaged on an inverted Olympus microscope (IX81), using a laser scanning confocal imaging system (Olympus Fluoview FV1000 with dual confocal scan heads) and an Olympus PlanApo 60× 1.42 $N_A$ objective. Rescuing complexin variants were C-terminally tagged with GFP separated by a 12 residue linker (GGSGGSGGSAAA), and synaptic protein levels were estimated by measuring background-subtracted fluorescence between dorsal cord synaptic peaks. Data were analyzed with custom software in IGOR Pro (WaveMetrics, Lake Oswego, OR; *Burbea et al., 2002*; Dittman and Kaplan, 2006). A fluorescent slide was imaged daily to monitor the laser stability and the dorsal cord fluorescence was normalized to the slide value.

## Electrophysiology

Whole-cell patch-clamp recordings were performed on dissected *C. elegans* as described previously (*Madison et al., 2005*; *McEwen et al., 2006*). Dissected worms were superfused in an extracellular solution containing 127 mM NaCl, 5 mM KCl, 26 mM $NaHCO_3$, 1.25 mM $NaH_2PO_4$, 20 mM glucose, 1 mM $CaCl_2$ and 4 mM $MgCl_2$, bubbled with 5% $CO_2$, 95% $O_2$ at 20°C. Whole-cell recordings were carried out at −60 mV using an internal solution containing 105 mM $CH_3O_3SCs$, 10 mM CsCl, 15 mM CsF, 4 mM $MgCl_2$, 5 mM EGTA, 0.25 mM $CaCl_2$, 10 mM HEPES and 4 mM $Na_2ATP$, adjusted to pH 7.2 using CsOH. Under these conditions we only observed cholinergic EPSCs. For low calcium experiments, 1 mM $CaCl_2$ was replaced with additional $MgCl_2$ for a total of 5 mM divalent cations.

## Protein purification

Protein expression constructs for NMR were cloned into a pET vector containing a $His_6$ tag and SUMO cleavage site to facilitate purification. A truncated polypeptide lacking the C-terminal domain (residues 1–77) was purified to simplify NMR and CD spectroscopic analysis. For NMR, BL21-DE3 *E. coli* cells were transformed and grown in Luria Broth (LB) containing 50 μg/ml Kanamycin to an optical density at 600 nm between 0.6 and 0.8. Cells were pelleted at 6500 rpm for 15 min, washed and resuspended for 30 min in a minimal media containing $^{15}N$ $NH_4Cl$, $^{13}C$ D-glucose, pelleted again, and resuspended in media containing $^{15}N$ $NH_4Cl$ and $^{13}C$ D-glucose prior to induction. To produce perdeuterated proteins, cells were grown directly in $D_2O$-based minimal media containing $^{15}N$ $NH_4Cl$, $^{13}C^2H$ D-glucose to an optical density at 600 nm of 0.6–0.8 prior to induction. Cells were induced with 400 μg/ml isopropyl thiogalactopyranoside (IPTG, OmniPur, Billerica, MA), and grown for three hours at 37°C. The cells were then pelleted, resuspended in lysis buffer (350 mM NaCl, 20 mM imidazole, 20 mM Tris pH 8, 1 mM EDTA, 0.1 mM PMSF, 1.7 mM BME, and 2 mM DTT) lysed by sonication, and pelleted at 40,000 rpm for 45 min. The supernatant was then bound to Ni-NTA beads and washed (wash buffer: 350 mM NaCl, 20 mM imidazole, 20 mM Tris pH 8, 1.7 mM BME, and 2 mM DTT). The protein was then eluted

(wash buffer with 250 mM imidazole), and fractions containing protein were combined and dialyzed overnight (dialysis buffer: 20 mM Tris pH 8, 150 mM NaCl, and 2 mM DTT). The dialyzed sample was incubated with a $His_6$-tagged SUMO protease to cleave the $His_6$ tag from CPX-1. The cleaved sample was incubated with Ni-NTA beads again to separate the protein from $His_6$–SUMO protease and the $His_6$–SUMO tag. The protein-containing fractions were eluted with wash buffer (as above), pooled, and dialyzed overnight into $ddH_2O$ and then lyophilized.

Protein expression constructs for CD were prepared as above with the following modifications: *E. coli* were grown to an optical density in LB + kanamycin at 600 nm between 0.6 and 0.8. Cells were then induced with IPTG, grown for four hours at 37°C, pelleted, and resuspended in lysis buffer (as above). Following sonication, the same procedures were followed as described above.

## NMR spectroscopy

Lyophilized protein samples were resuspended to a final concentration of 50–75 µM in 100 mM NaCl, 50 mM PIPES pH 6.08, and proton-nitrogen (HSQC) spectra as well as a standard set of heteronuclear triple resonance three-dimensional spectra were collected on a Varian Unity Inova 600 MHz (Weill Cornell NMR Facility) spectrometer equipped with a cryoprobe and additional 3D and 4D (H)N(COCA)NNH spectra were collected using a perdeuterated full-length construct on a Bruker Avance 800 MHz spectrometer with cryoprobe (New York Structural Biology Center, NY). Previous backbone resonance assignments for full-length free wild-type complexin were confirmed and extended to achieve a completeness level of 93%. Backbone resonance assignments for free R43P complexin were obtained at a completeness level 84%. All spectra were collected at 20°C. Data were processed using NMRpipe and analyzed using NMRview (*Johnson and Blevins, 1994*; *Delaglio et al., 1995*). Spectra were referenced indirectly to water. Protein concentrations were measured using a Bradford reagent (Bio-Rad, Hercules, CA).

## CD spectroscopy

Lyophilized protein samples were resuspended in 40 mM phosphate buffer containing 100 mM NaCl to between 1 and 2 mM. CD spectra were then taken from 250 nm to 200 nm on both Aviv 62DS and Aviv Model 410 instruments at 25°C. The 2,2,2- trifluoroethanol (TFE) experiments were performed by adding increasing volumes of TFE (JT Baker) added to the samples. Data were background subtracted, and averages of two sequential scans were computed. The process was repeated and averaged to generate the data displayed in *Figures 4,6*. Percentage helicity from CPX-1 spectra was estimated by calculating the best fit to a linear combination of pure helix and pure random coil spectra (*Saxena and Wetlaufer, 1971*). The percent helicity was then plotted against the percentage of TFE used, and fits were obtained to the following equation:

$$h = h_{min} + \frac{(h_{max} - h_{min})}{1 + (K/[TFE])^n}$$

where $h$ is the predicted % helicity, $h_{min}$ is the minimum value of helicity observed, at 0% TFE, $h_{max}$ is the maximum value of helicity, $K$ is the dissociation constant for the equilibrium between the random coil + TFE and the alpha-helix conformation, [TFE] is the percentage of TFE, and $n$ is the cooperativity. The $h_{min}$ value was allowed to change for each protein, but the $h_{max}$ value was kept constant at the wild-type protein's value. The 7-turn artificial helix construct possesses an intrinsically larger helical percentage at high [TFE], so $h_{max}$ was not held constant when fitting the TFE titration for this peptide. $K$ and $n$ were found using least squares minimization.

## Prediction of protein secondary structure

Protein sequences were entered into the online helical prediction software, Agadir, http://agadir.crg.es/ using default settings (*Munoz and Serrano, 1997*). To calculate cross-phylogeny comparisons in *Figure 1*, the full length protein was entered into Agadir, and the 18 amino acids N-terminal to the beginning of the central helix were averaged. To calculate the percent helicity and Cα shifts, only the residues corresponding to the protein constructs were entered into Agadir. Hydrophobic moments were calculated using an online tool available at http://rzlab.ucr.edu/scripts/wheel/wheel.cgi using standard interface hydrophobicity values for each residue (*Wimley and White, 1996*; *Wimley et al., 1996*). To calculate charge density across the AH amino acids predicted to be at least 5% helical by Agadir, the number of negatively charged amino acids was subtracted from the number of positively charged amino acids and the difference was divided by the total number of residues.

## Statistical analysis

For all datasets in this study, statistical comparisons were made across the entire dataset using the Tukey–Kramer method for multiple comparisons with p < 0.01 as the significant criterion. Some of this data was then used in multiple figures. Average values for wild-type and *cpx-1* mutant voltage-clamp recordings from the same dataset were shown in *Figures 2–4* and *Figure 6*. The aldicarb time course data for wild-type, *cpx-1*, and full-length CPX-1 rescue animals is used in *Figures 2–6*. Likewise, the aldicarb rescue data for ΔAH$_{short}$ is displayed in *Figures 2,4,5* while the R43P rescue data is used in *Figures 4,5*. The wild-type CPX-1(ΔCT) NMR chemical shift data are used in both *Figure 1* and *Figure 4*.

## Acknowledgements

We thank Tim Ryan, Mark Bowen, Keith Weninger, Rebecca Davidson, and members of the Dittman lab for advice, discussions, and manuscript suggestions. We thank the HTSRC at Rockefeller University for CD spectroscopy support. This work was supported by National Institutes of Health grants R01-GM095674 (JSD), R01-NS085214 (JB), R37-AG019391 (DE), a gift from Herbert and Ann Siegel (DE), F30-MH101982 (DS), MSTP T32-GM007739 (DS), and the Rita Allen Foundation Award 188423 (JSD). DE is a member of the New York Structural Biology Center. The data collected at NYSBC was made possible by a grant from NYSTAR and ORIP/NIH facility improvement grant CO6RR015495. The 900 MHz NMR spectrometers were purchased with funds from NIH grant P41GM066354, the Keck Foundation, New York State Assembly, and U.S. Dept. of Defense.

## Additional information

### Funding

| Funder | Grant reference number | Author |
| --- | --- | --- |
| National Institute of General Medical Sciences | R01-GM095674 | Jeremy S Dittman |
| Rita Allen Foundation | 188423 | Jeremy S Dittman |
| National Institute on Aging | R37-AG019391 | David Eliezer |
| National Institute of Neurological Disorders and Stroke | R01-NS085214 | Jihong Bai |

The funders had no role in study design, data collection and interpretation, or the decision to submit the work for publication.

### Author contributions

DTR, Conception and design, Acquisition of data, Analysis and interpretation of data, Drafting or revising the article; YD, DS, Acquisition of data, Analysis and interpretation of data; JB, Conception and design; DE, JSD, Conception and design, Analysis and interpretation of data, Drafting or revising the article

### Author ORCIDs

Yongming Dong, http://orcid.org/0000-0002-6510-0913
David Eliezer, http://orcid.org/0000-0002-1311-7537

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
