## [Decision Letter]

Thank you for sending your work entitled “Complexin's accessory helix functions by stabilizing central helix secondary structure” for consideration at *eLife*. Your article has been favorably evaluated by Eve Marder (Senior editor), a Reviewing editor, and 2 reviewers. We are very happy to say that your work has been accepted for publication in *eLife*.

This is a nice study in which the authors address the role of the accessory helix in the protein complexin. Previous work showed that the central helix mediates binding to the surface of the neuronal SNARE complex, and this interaction is essential for function in regulated exocytosis. Numerous studies have also shown that the accessory helix, localized N-terminally directly adjacent to the central helix, is also required for function but the mechanism is controversial. In particular, the discussion is dominated by a model that suggests a direct competitive binding of this helix to a partially assembled SNARE complex, preventing zippering of synaptobrevin. Here the authors use an elegant combination of genetic rescue experiments in *C. elegans* and structural methods to show that for function of the accessory helix all that is required is a helical structure of this domain, with no requirement for a specific sequence, charge density, or overall hydrophobicity. The experiments are elegant and compelling, and in my opinion the work settles a rather long-standing controversy in the field. The authors are very cautious in their Discussion (evidently, they are aware of strongly opinionated colleagues in this field), but I would like to encourage them to be more explicit. The data and conclusions make it highly unlikely that the accessory helix functions by another specific protein-protein interaction. The findings will make a splash as they challenge and dismiss, one-by-one, essentially all theories on complexin function that have been put forward so far (hydrophobic/electrostatic protein interactions...). The authors are to be congratulated on such a nice study.

[Editors’ note: minor issues and corrections have not been included, so there is not an accompanying Author response.]